# Novel Biological and Molecular Characterization in Radiopharmaceutical Preclinical Design

**DOI:** 10.3390/jcm10214850

**Published:** 2021-10-21

**Authors:** Nicoletta Urbano, Manuel Scimeca, Anna Tolomeo, Vincenzo Dimiccoli, Elena Bonanno, Orazio Schillaci

**Affiliations:** 1Nuclear Medicine Unit, Department of Oncohaematology, Policlinico “Tor Vergata”, 00133 Rome, Italy; n.urbano@virgilio.it; 2Department of Experimental Medicine, University of Rome “Tor Vergata”, 00133 Rome, Italy; elena.bonanno@uniroma2.it; 3San Raffaele Roma Open University, 00166 Rome, Italy; 4Saint Camillus International University of Health Sciences, 00131 Rome, Italy; 5Department of ITELPHARMA, ITEL Telecomunicazioni S.R.L., Via Labriola snc, 70037 Ruvo di Puglia, Italy; a.tolomeo@itelte.it (A.T.); v.dimiccoli@itelte.it (V.D.); 6Department of Biomedicine and Prevention, University of Rome “Tor Vergata”, 00133 Rome, Italy; orazio.schillaci@uniroma2.it; 7IRCCS Neuromed, 86077 Pozzilli, Italy

**Keywords:** nuclear imaging, pre-clinical model, radiopharmaceutical, digital autoradiography system

## Abstract

In this study, the potential of a digital autoradiography system equipped with a super resolution screen has been evaluated to investigate the biodistribution of a 18F-PSMA inhibitor in a prostate cancer mouse model. Twelve double xenograft NOD/SCID mice (LNCAP and PC3 tumours) were divided into three groups according to post-injection time points of an 18F-PSMA inhibitor. Groups of 4 mice were used to evaluate the biodistribution of the radiopharmaceutical after 30-, 60- and 120-min post-injection. Data here reported demonstrated that the digital autoradiography system is suitable to analyse the biodistribution of an 18F-PSMA inhibitor in both whole small-animal bodies and in single organs. The exposure of both whole mouse bodies and organs on the super resolution screen surface allowed the radioactivity of the PSMA inhibitor distributed in the tissues to be detected and quantified. Data obtained by using a digital autoradiography system were in line with the values detected by the activity calibrator. In addition, the image obtained from the super resolution screen allowed a perfect overlap with the tumour images achieved under the optical microscope. In conclusion, biodistribution studies performed by the autoradiography system allow the microscopical modifications induced by therapeutic radiopharmaceuticals to be studied by comparing the molecular imaging and histopathological data at the sub-cellular level.

## 1. Introduction

In the era of 4P medicine (predictive, preventative, personalized, participatory), the development of new therapies and/or diagnostic procedures requires continuous and constant enhancement of the technological armamentarium available for researchers. In this context, multidisciplinary approaches offer the chance to identify and develop new molecules for personalized target therapies [1,2,3].

Molecular imaging investigations, both in pre-clinical models and clinical trials, in collaboration with other biomedical disciplines, such as histology, pathology and molecular biology, currently represent a scientific multidisciplinary platform for developing appropriate pre-clinical models more and more similar to the complex mechanisms of human diseases [4,5,6]. Molecular imaging procedures can be used to assess patients’ state of health earlier and to carefully choose the best clinical individual design for each single patient by translational applications to reach a novel methodological approach (multimodal, theragnostic, pre-targeting) that aims at a deep understanding of pathologies [7].

In recent years, several new molecules have been proposed as radiopharmaceuticals; however, only a small percentage of them have reached the criteria for their employment in clinical practice. This fact frequently occurs due to the lack of a detailed characterization of pre-clinical models, as well as the absence of the necessary technology to achieve these purposes.

Macroscopic visualization of cellular mechanisms by dedicated positron emission tomography (PET) in pre-clinical research certainly represents a very appreciated upcoming imaging technology, even if its high cost limits its access to the research community. Thus, numerous promising molecules showing interesting in vitro data about their affinity to their ligands are not furtherly investigated for lack of funds or dedicated micro-molecular imaging devices [8]. 

Therefore, the development of appropriate and pivotal methodologies capable of supporting researchers in radiopharmaceutical pre-clinical studies and the possibility to combine imaging diagnostic data with histopathology and/or molecular biological analysis could provide a crucial incentive for developing biomedical research involved in the realization of tailored target therapies. 

In this scenario, we adopted a digital autoradiography system comprised of a laser scanning device (Cyclone Plus PerkinElmer, Inc., Waltham, MA, USA) commonly used for radiopharmaceutical thin layer chromatography (TLC) quality control in nuclear pharmacy practice to characterize quantitative imaging of spatial radioactivity distribution on animal tissue sections to develop the best procedure for a new radiopharmaceutical scale-up while assuring the best safety route in a pre-clinical development package [9]. The use of this autoradiography system with a super resolution (SR) storage phosphor screen allows the distribution of the radioactivity to be examined as well as the amount of detected radioactivity in terms of Digital Light Units (DLU) to be quantified very quickly on both the whole animal (mouse, rat) and excised organs [9]. 

Moreover, the ability to closely associate biodistribution data with histopathological images in animal models could enable the characterization of investigated molecular and sub-molecular events crucial for the implementation of personalized medicine, especially in malignant neoplasms such as prostate cancer (PC). 

In this research, a mouse model of PC was used to investigate the biodistribution of a new molecular compound labelled with Fluorine-18 (18F) capable of selectively binding the PSMA (prostate-specific membrane antigen) expressed by PC cells. 18F-PSMA inhibitor was studied within xenograft tumours and mouse organs by using a digital autoradiography system, an activity calibrator (Talete, Comecer, Castel Bolognese, Italy) and histopathological investigation to analyse aspects of pharmacodynamics, pharmacokinetics and toxicology at the sub-cellular level.

This proposed design could open new and interesting perspectives in molecular precision medicine, also representing a valid strength in theragnostics, i.e., in targeted cancer medicine through molecular radiotherapy.

In fact, the ability to image, quantify and characterize radionuclides with different emission properties, as well as β- and α-particles, enables activity distribution at microscopic scale of therapeutic compounds to be resolved, ensuring a biological response and toxicity prediction in dosimetry analysis of all radiopharmaceutical studies [10,11].

## 2. Materials and Methods

All pre-clinical studies must comply with the guidelines released by Good Laboratory Practices in Nuclear Medicine and Anatomic Pathology Departments with ISO (International Organization for Standardization) certification.

In order to evaluate the possible use of the autoradiography system, 12 NOD/SCID mice with xenografts of 4 to 6 mm in diameter of both LNCAP and PC3 tumours were used. Mice were divided into 3 groups according to post-injection time points of 18F-PSMA inhibitor. In particular, 4 NOD/SCID xenograft mice were used to evaluate the biodistribution of the radiopharmaceutical at 30 min post-injection (Group 1), 4 NOD/SCID xenograft mice were used to evaluate the biodistribution of the radiopharmaceutical at 60 min post-injection (Group 2) and 4 NOD/SCID xenograft mice were used to evaluate the biodistribution of the radiopharmaceutical after 120 min (Group 3). Xenografts with LNCAP cells (PSMA positive) were used to investigate the in vivo affinity of the 18F-PSMA inhibitor for its biological target. PC3 tumours were used as control.

### 2.1. Cell Lines

Both PSMA-expressing (LNCAP) and PSMA-non-expressing (PC3) prostate cancer cell lines were grown in RPMI 1640 medium (Invitrogen, Carlsbad, CA) containing 10% fetal bovine serum (FBS) (Invitrogen) and 1% Pen-Strep (Biofluids, Camarillo, CA, USA). All cell cultures were maintained in 5% carbon dioxide (CO_2_) at 37.0 °C in a humidified incubator.

### 2.2. Cell Culture Immunoflurescence

Immunofluorescence investigations were performed to verify the PSMA expression and proliferation index (ki67 expression) of both LNCAP and PC3 prostate cancer cell lines.

Cells were plated on poly-l-lysine coated slides (Sigma-Aldrich cat #P4707) in 24-well cell culture plates and fixed in 4% paraformaldehyde. After pre-treatment with EDTA citrate at 95 °C for 20 min and 0.1% Triton X-100 for 15 min, cells were incubated 1 h with the mouse monoclonal anti-PSMA antibody (rabbit monoclonal clone SP29; Ventana, Tucson, AZ, USA) and rabbit monoclonal anti-Ki67 antibody (rabbit monoclonal clone 30-9; Ventana, Tucson, AZ, USA). Washings were performed with PBS/Tween20, pH 7.6. Reactions were revealed by using FITC-goat anti-mouse secondary antibodies (Novus Biologicals, Littleton, CO, USA) for PSMA and Texas red goat anti-rabbit secondary antibodies (Novus Biologicals, Littleton, CO, USA) for Ki67. DAPI (Novus Biologicals, Littleton, CO, USA) was used to stain the nucleus (see Appendix A).

### 2.3. Animal Model

Five- to six-week-old male, non-obese diabetic (NOD)/severe combined immunodeficient (SCID) mice (*n* = 20) (ENVIGO, Huntingdon, UK) were implanted subcutaneously (s.c.) with LNCAP and PC3 cells (2 × 106 in 100 µL of Matrigel) at the forward left and right flanks, respectively. Mice were kept in a temperature-controlled room (25 ± 2 °C) at 50% relative humidity with a 12/12 h light/dark cycle and had ad libitum access to food and water. Mice were used in ex vivo biodistribution assays when the xenografts reached 4 to 6 mm in diameter. Tumour growth was monitored daily by measuring tumour mass in two dimensions with a digital caliper. Tumor volumes were calculated according to the following formula: tumor volume (mm 3) = [length (mm) × width 2 (mm 2)]/2.

### 2.4. Biodistribution Study

All mice were injected with 3,70 MBq of a 18F-PSMA inhibitor via the lateral tail vein.

For each group, one mouse was used to study the biodistribution across the whole-body, while the biodistribution of the remaining three mice was evaluated on the main organs after post-mortem excision (Figure 1). The radioactivity in the organs, tumours, hearth, lung, kidneys, bowel and liver was detected both in terms of DLU using a SR storage screen by autoradiography system, and, immediately after measurements, in terms of MBq by using an activity calibrator considering radioactive decay.

For whole-body evaluation, radioactive sacrificed mice were exposed to the SR storage screen, protected by a transparent film, for 10 min. Excised organs were exposed in the same way.

According to previous tests, the optimum exposure time (10 min) was adopted considering, above all, the small amount of radioactivity (about 3–4 MBq) in our samples and the need to have the best picture quality due to the increased amount of data in the image without saturation of the phosphor layer.

Subsequently, the exposed SR storage phosphor screen was scanned at 150 DPI (170 mm pixel size) to create a digitized image for analysis. Scanning took 3 min only.

The SR screen was re-usable after erasing it by exposure to UV-free white light, such as a white light translumitor used to examine X-ray films, for a few mins.

### 2.5. Histology

At the end of the biodistribution studies, organs were fixed in formalin for 24 h and paraffin-embedded [12]. Three-µm serial sections were haematoxylin-eosin stained and used for both morphological study and mitosis counts.

### 2.6. Immunohistochemistry

Three-µm paraffin serial sections were used to evaluate the expression of a prognostic biomarker, vimentin. Specifically, antigen retrieval was performed on 3 μm thick paraffin sections using EDTA citrate pH 7.8 for 20 min at 95 °C. Sections were then incubated for 30 min at room temperature with the following primary antibodies: pre-diluted anti-Ki67 (Rabbit monoclonal clone 30-9; Ventana, Tucson, AZ, USA), pre-diluted anti-PSMA (Rabbit monoclonal clone SP29; Ventana, Tucson, AZ, USA) and anti-vimentin (mouse monoclonal clone V9; Ventana, Tucson, AZ, USA). Washings were performed with PBS/Tween20 pH 7.6. Reactions were revealed by the HRP-DAB Detection Kit (UCS Diagnostic, Rome, Italy). Immunohistochemical reactions were evaluated by counting the number of positive prostate cells on 500 in total in randomly selected regions.

### 2.7. Transmission Electron Microscopy

Ultrastructural investigations were performed by transmission electron microscopy to study the ultrastructure characteristics of xenograft tumours after biodistribution analysis. Small fragments of each tumour were treated as previously described [13].

Briefly, one millimeter^3^ of tissue from each tumour specimen was fixed in 4% para formaldehyde (PFA) and post-fixed in 2% osmium tetroxide [13]. After washing with 0.1 M phosphate buffer, the sample was dehydrated by a series of incubations in 30%, 50%, and 70%, ethanol. Dehydration was continued by incubation steps in 95% ethanol, absolute ethanol, and propylene oxide; then, samples were embedded in Epon (Agar Scientific, Stansted Essex, UK). Eighty µm ultra-thin sections were mounted on copper grids and observed with a Morgagni FEI transmission electron microscope (FEI Company, Hillsboro, OR, USA).

### 2.8. Statistical Analysis

One-way ANOVA and Mann–Whitney tests will be used to investigate the data of the biodistribution investigations (both DLU and MBq).

## 3. Results

### 3.1. Cell Cultures Characterization

Immunofluorescence analysis was performed to characterize the cell cultures before the xenograft development. Specifically, after confluence, each cell line was tested for the expression of PSMA and Ki67 (proliferation mark). As one aspect, the LNCAP cell line was characterized by more than 90% of PSMA-positive cells. Conversely, no/rare PSMA positive cells were observed in PC3 cells. Similar expression of Ki67 was observed in both LNCAP and PC3 cell lines.

### 3.2. Measurement Evaluation by Activity Calibrator

A significant increase in radioactivity was detected in LNCAP tumours (PSMA positive) (0.637 ± 0.11 MBq) as compared to both PC3 tumours (PSMA negative) (0.234 ± 0.08 MBq) (Figure 2A) and all other examined organs after 30 min (Figure 2B).

High values of radioactivity, though significantly lower with respect to LNCAP tumours, were observed in kidneys 0.351 ± 0.15 MBq). Of note, a constant increase of the uptake of the radiopharmaceutical was observed in tumours expressing the biological target (LNCAP) (60 min 0.703 ± 0.57 MBq; 120 min 0.772 ± 0.98 MBq). Otherwise, a constant decrease of the radioactivity was observed in the remaining organs after both 60 and 120 min (see Table 1). This condition is in line with the decay of the radioisotope (18F). All measurements were corrected for radioactive decay.

### 3.3. Radioactivity Detection by Digital Autoradiography System

The analysis performed by the autoradiography system allowed the biodistribution study of 18F-PSMA inhibitor (Figure 3A–D) to be performed. In particular, after 30 min, 18F-PSMA inhibitor was widespread throughout the animal (Figure 3D), although the analysis of individual organs showed a significant increase in the uptake of the radiopharmaceutical in LNCAP tumours (Figure 3D).

It is noteworthy that, 60 min after the injection, the evaluation of radioactivity in the mouse displayed uptake for LNCAP tumours and bladder (Table 2). At 120 min, the autoradiographic analysis was able to show the uptake of the radiolabelled PSMA inhibitor only in the LNCAP tumours (Table 2). DLU data showed the same trend of radioactivity value detected by the activity calibrator (Table 2 and Figure 3A,B). The proportional relationship between DLU and MBq was assayed in our previous in vitro experiments. In a complex biological contest like this, DLU data, however, demonstrated results in line with the activity calibrator measurement.

Both investigations showed a progressive increase of the radioactivity in LNCAP tumours as well as a constant reduction of the radioactivity in all other investigated organs, including PC3 tumours.

### 3.4. Histological and Immunohistochemical Analysis

The histological investigations showed no significant morphological alterations both in tumours and other organs. Comparative analysis between histological and autoradiographic images of LNCAP-positive tumours displayed a strictly spatio-temporal association between the uptake of 18F-PSMA inhibitor and the presence of mitotic figures (Figure 3E). Specifically, areas with higher radiopharmaceutical uptake were characterized by the presence of several mitoses (Figure 3E). Similarly, a spatio-temporal association was observed comparing the autoradiographic images with the expression of Ki67 (Figure 3E). Immunohistochemical evaluation of PSMA confirmed the expression of this molecule only in LNCAP xenografts (Figure 4A,B). High numbers of Ki67 positive cells were observed in both PC3 and LNCAP xenografts (Figure 4C,D). Additionally, the number of vimentin positive cells was ≥50% in each xenograft tissue, thus demonstrating a similar level of tumour differentiation (Figure 4E,F). Indeed, vimentin filaments are expressed by undifferentiated prostate cancer cells.

These preliminary data support the idea that the high-resolution filmless autoradiography phosphor imager can be useful to perform comparative studies in which biodistribution of a radiopharmaceutical is associated with histological images at the sub-cellular level.

### 3.5. Electron Microscopy

Transmission electron microscopy analysis of xenograft tumours (both LNCAP and PC3) displayed heterogeneous epithelial cancer populations (Figure 5). Specifically, both well-differentiated and mesenchymal-like cells were observed (Figure 5A–D). However, PC3 tumour mass (Figure 5C,D) was characterized by a higher number of mesenchymal-like cells with respect to LNCAP (Figure 5A,B). In addition, mitotic figures were often detected. No/rare apoptotic cells were found. After biodistribution studies, a moderate increase in apoptotic cells were noted in LNCAP xenograft tumours with respect to PC3.

## 4. Discussion

Small-animal imaging has become a fundamental technique for the development of new diagnostic or therapeutical radiopharmaceuticals. Indeed, currently, pre-clinical imaging of animal models represents an invaluable tool in studying the etiopathogenesis of and therapeutic responses in various human pathologies such as neurological, cardiovascular and oncological diseases [14]. Molecular imaging techniques can be used to assess biological processes at the cellular and molecular levels, enabling the detection of disease in very early or pre-symptomatic stages, and to estimate the efficacy of novel therapies in individual patients [15,16,17,18]. The assessment of biological properties of tumours, such as metabolism, proliferation, hypoxia, angiogenesis, apoptosis, and gene and receptor expression, contributes to the realization of precision medicine [19,20], owing to the possibility of monitoring physio-pathological processes in vivo, detecting therapeutic responses, identifying non-responders at an early stage, and enabling the switch to novel therapeutic approaches [21,22]. In this context, PC represents a unique model for the realization of new protocols of personalized medicine. Indeed, PC is a very heterogeneous disease, and contemporary management is focused on identification and treatment of the prognostically adverse high-risk tumours while minimizing overtreatment of indolent, low-risk ones [23]. In recent years, imaging has gained increasing importance in the detection, staging, posttreatment assessment and detection of recurrence of PC [24,25,26]. Several imaging modalities, including conventional and functional methods, are used in different clinical scenarios with their very own advantages and limitations. Thus, several groups are involved in the development of new radiopharmaceuticals for both the diagnosis and therapy of PC. To these aims, some laboratories now have a combination of different small-animal imaging systems, which are being used by biologists, pharmacists, physicians and physicists.

Unfortunately, the number of laboratories equipped with innovative small-animal imaging systems are currently very few, due to the high costs of these scientific devices. This fact often precludes the development of several promising radiopharmaceuticals. Thus, the enhancement of an instrumental armamentarium available for researchers could significantly increase the chance of success of pre-clinical investigations based on the identification of new radiolabelled molecules.

For several years, the in situ detection of radiolabeled molecules has been performed by using film or film emulsion (conventional autoradiographic analysis). Despite the fact that the spatial resolution obtained with these devices is very good, the sensitivity of film for low activity levels is poor, due to the low x-ray/β particle detection efficiency. According to this, film autoradiographs frequently must take several days to produce a satisfactory image. In addition, the limited dynamic range of film can cause under- or over-exposure of parts of the image. Therefore, better autoradiography systems based on digital position-sensitive detectors have been developed. Among these, the most sensitive are phosphor imaging plates [27], multiwire proportional chambers [28], scintillating optical fibres [29], microchannel plates [30], silicon strip detectors [31], and silicon or gallium arsenide pixel detectors [32]. Moreover, in the last years, extremely sensitive digital autoradiographs have been developed both for quality control and in vivo research.

Starting from these considerations, in this study, the potential of a digital autoradiography system equipped with an SR screen has been evaluated to characterize 18F-PSMA inhibitor biodistribution in a PC mouse model within xenograft tumours and mouse organs. In addition, a multidisciplinary investigation including histopathological analysis was performed to study radiopharmaceutical behavior at the sub-cellular level.

A digital autoradiography system is a very versatile and sensitive device for radioisotope imaging, replacing film autoradiography [33]. This system has been designed for a great variety of applications, such as the analysis of purity for radiopharmaceuticals, nucleotide metabolism studies, in vitro imaging of tissue sections and also gene and protein expression studies [33]. In fact, it can image and quantify activity distribution of different radionuclides (photon-, β- and α-particles emitting).

An SR phosphor screen is a flexible support film formulated with the finest grade of barium fluorobromide and containing traces of bivalent europium (BaFBr/Eu2+) phosphor crystals, which acts as a bioluminescence center to provide the best resolution. When the screen is exposed to a radioactive sample, the energy of the radioisotope ionizes the Eu+ 3 to Eu2+, liberating electrons which are trapped in the bromine vacancies [34]. Subsequently, the exposed SR screen, wrapped around the carousel of the photometer reading device, is scanned by a focused red light laser beam (633 nm); the laser-stimulated luminescence releases blue light photons (390 nm) which are detected by a photo-multiplier tube (PMT) and converted to electrical signals expressed as DLU. The SR screen was scanned in a few minutes to create a high-resolution digitized image of the locations and intensity of the radioactivity in the sample, which is quantified by OptiQuantTM image analysis software and stored for future reference.

The data here reported showed that the digital autoradiography system is suitable to analyse the biodistribution of an 18F-PSMA inhibitor in both whole small-animal bodies (mice) and in single organs. Specifically, the exposure of both whole mouse bodies and organs on the SR screen surface allowed the radioactivity of the PSMA inhibitor distributed in the tissues to be detected and quantified. It is noteworthy that data obtained by using the digital autoradiography system were in line with the value of measurement detected by the activity calibrator, thus highlighting the high sensitivity of the digital autoradiography system. As expected, a significant and constant increase in the uptake of PSMA inhibitor was observed only in PSMA-positive tumours (LNCAP), while a decrease in the value of radioactivity was noted in other investigated organs as well as in the PSMA negative-tumours (PC3). These data were supported by the immunophenotypical characterization performed on both prostate cancer cell cultures and xenograft tumours. Indeed, no/rare PSMA-positive prostate cancer cells were observed.

Even though the distribution of radioactivity evaluated on whole mouse bodies by the digital autoradiography system cannot have the same sensitivity of micro-PET investigation, it allows an excellent space-time assessment of the biodistribution of a radiopharmaceutical. In particular, in this study, it was possible to follow the biodistribution of a PSMA inhibitor at three different time points, observing a progressive increase of radioactivity in the PSMA-positive tumour area.

If compared with micro-PET investigations, the main limitation of the use of a digital autoradiography system equipped with an SR screen for biodistribution studies is the impossibility to perform the analysis on live animals. In fact, autoradiographic investigation needs a non-dynamic system to obtain a high-resolution image of the radioactivity of the tissues. Conversely, micro-PET investigations can also provide physiological and pharmacological process quantitation by dynamic acquisitions of fast kinetic data [35,36]. However, the use of micro-PET devices does not increase the animal’s welfare. Indeed, according to the D.L. 4 March 2014, No. 26; directive 2010/63/EU of the European parliament and council “Guide for the Care and Use of Laboratory Animals, United States National Research Council, 2011”, animals must be sacrificed at the end of the experimental phase for studies using both micro-PETs and this design. The difference only concerns the possibility to perform dynamic acquisitions. Micro-PET devices are extremely sensitive, having the capability to perform detections at a picomolar level, but only with PET radiopharmaceuticals. In fact, therapeutic radiopharmaceuticals cannot be detected by micro-PET, limiting the possibility of performing accurate theragnostic investigations [37,38].

From our point of view, a further possible weakness for micro-PET investigations is the low spatial resolution offered by dedicated devices. This limitation makes it difficult to perform accurate comparison analysis between the radiopharmaceutical uptake and microscopic characteristics of tissues. In fact, it is impossible to overlap the 3D micro-PET images with the histological characteristics of investigated tissues.

The use of a digital autoradiography system equipped with an SR screen can represent an extraordinary and less expensive alternative for all research groups to perform preliminary and multidisciplinary imaging investigations with all available radiopharmaceuticals. It is important to note that the methodology proposed here could also be associated with the analysis of whole-body slices performed by micro-PET investigations to delineate more precise localization of the radiotracer.

In this study, the use of the SR screen makes it possible to associate the uptake of the PSMA inhibitor with the microscopic characteristics of the investigated tumours. Remarkably, the high-resolution image obtained from the SR screen allowed for a perfect overlap with the tumour image achieved under the optical microscope, showing an association between the uptake of PSMA inhibitor and the proliferation index of the tumours (number of mitoses). This type of analysis was possible due to the bi-dimensional characteristics of both images. Indeed, histological evaluation has been performed on 3 µm-thick paraffin sections obtained no more than 50 microns away from the surface of the tumour that impressed the SR screen. Similarly, serial paraffin sections were used to compare the uptake images with immunohistochemical ones (Ki67m PSMA and vimentin), opening the way for comparison studies in which radiopharmaceutical uptake is associated with in situ molecular data. This analysis could be useful to investigate the molecular events related to the biodistribution of radiopharmaceuticals in tissues and cells. In addition, small fragments of xenograft tumours investigated by electron microscopy could provide subcellular information capable of explaining the possible mechanisms related to the radiopharmaceutical’s uptake, as well as the ultrastructural modifications induced by the radiopharmaceuticals.

Therefore, the autoradiographic analysis, in addition to the lower management costs, shows peculiar characteristics that make it a powerful device for biodistribution studies as an alternative or support for micro-PET investigations. Notably, the inability of this autoradiographic system to detect different radioisotopes makes it possible to use this device to develop specific and detailed theragnostic in vivo characterizations also implementing dosimetry analysis in biological response and toxicity predictions of all radiopharmaceutical studies.

Even more importantly, biodistribution studies performed by the autoradiography system may allow the microscopical modifications induced by therapeutic radiopharmaceuticals to be studied by comparing the molecular imaging and histopathological data at the sub-cellular level. However, it has not escaped our notice that there is need to perform further studies focused on the biodistribution of diagnostic emitters (gamma, X) and/or therapeutic radionuclides in order to complete the validation of the digital autoradiography system here described. 

## 5. Conclusions

In conclusion, a crucial driver for future advances in nuclear imaging, and more generally, in personalized medicine is the availability of devices for biodistribution studies which can be easily purchased and used by research institutes around the world. The strong translational science potential of biodistribution studies of new radiopharmaceuticals in small animal models holds great promise to dramatically advance our understanding of human disease. The assessment of molecular and functional processes using imaging agents as either direct or surrogate biomarkers will ultimately enable the characterization of disease expression in individual patients and thus facilitate tailored treatment plans that can be monitored for their effectiveness in each subject.

These considerations further emphasize the importance of pre-clinical studies in biomedical research, promoting every effort to prevent promising scientific investigations from being discontinued due to lack of funds or specific biomedical devices. The incapacity to investigate in vivo a promising molecule for the treatment or diagnosis of human disease should be considered a debacle for the scientific community. Therefore, this preclinical design could bridge this gap by ensuring a better comprehension of the complexity of mechanisms of human disease.

## Figures and Tables

**Figure 1 jcm-10-04850-f001:**
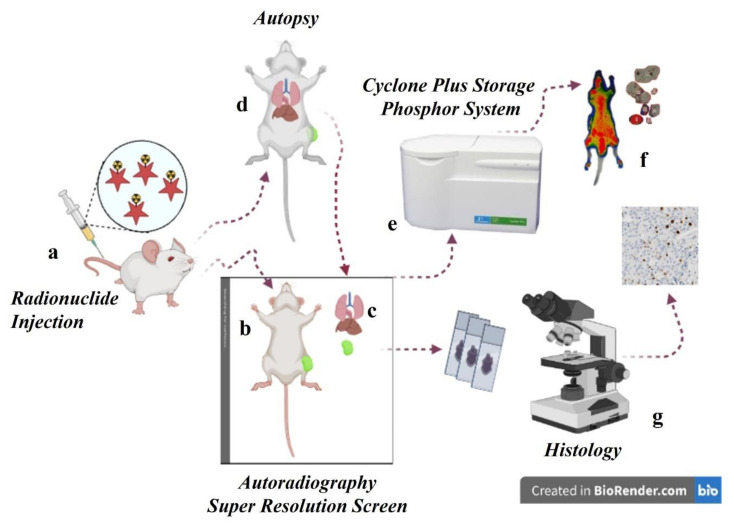
Representative scheme of the biodistribution study with the digital autoradiographic system. After the radionuclide injection (**a**) mice are used for both whole-body imaging (**b**) and single organ analysis (**c**) after autoptic examination (**d**). The exposed super resolution storage phosphor screen is then scanned at 150 DPI (170 mm pixel size) (**e**) to create a digitized image for analysis (**f**). Simultaneously, excised organs can be used to perform histological analysis (**g**).

**Figure 2 jcm-10-04850-f002:**
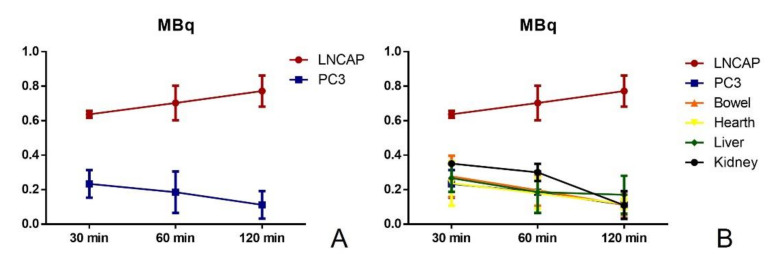
Measurement by activity calibrator. (**A**) Graph shows the radioactivity value detected by the activity calibrator in terms of MBq in LNCAP and PC3 tumours after 30, 60 and 90 min. (**B**) Graph displays the radioactivity value detected by the activity calibrator in terms of MBq in LNCAP, PC3, bowel, heart, liver and kidney tumours after 30, 60 and 90 min.

**Figure 3 jcm-10-04850-f003:**
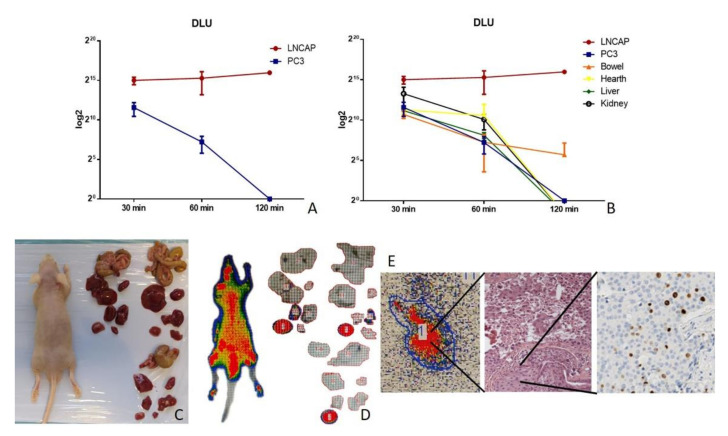
Evaluation of radioactivity detection by a digital autoradiography system and histological analysis. (**A**) Graph shows the radioactivity value detected by a digital autoradiography system in terms of DLU in LNCAP and PC3 tumours after 30, 60 and 90 min. (**B**) Graph displays the radioactivity value detected by a digital autoradiography system in terms of DLU in LNCAP, PC3, bowel, heart, liver and kidney tumours after 30, 60 and 90 min. (**C**) Whole body and excised organs. (**D**) Autoradiographic image shows 18F-PSMA inhibitor uptake in both the whole body and excised organs. (**E**) Morphological and immunohistochemical images of LNCAP tumours reveal an association among 18F-PSMA inhibitor uptake, mitosis and ki67 expression.

**Figure 4 jcm-10-04850-f004:**
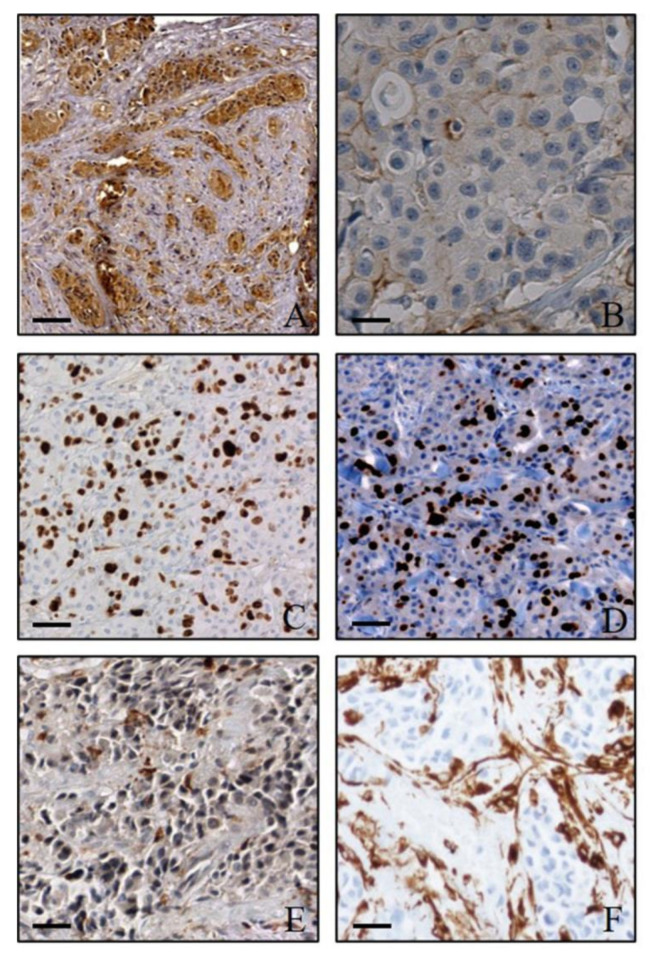
Immunohistochemical investigation of xenograft tumours. (**A**) LNCAP xenograft tumour mass characterized by numerous PSMA-positive cells. (**B**) No/rare PSMA-positive cells in PC3 tumour mass. (**C**,**D**) Images show numerous Ki67 positive cells in both LNCAP (**C**) and PC3 (**D**) xenografts. (**E**) Moderate expression of vimentin in a LNCAP xenograft tumour. (**F**) Image displays numerous vimentin-positive cells in a PC3 xenograft. Scale bar 100 µm for all images.

**Figure 5 jcm-10-04850-f005:**
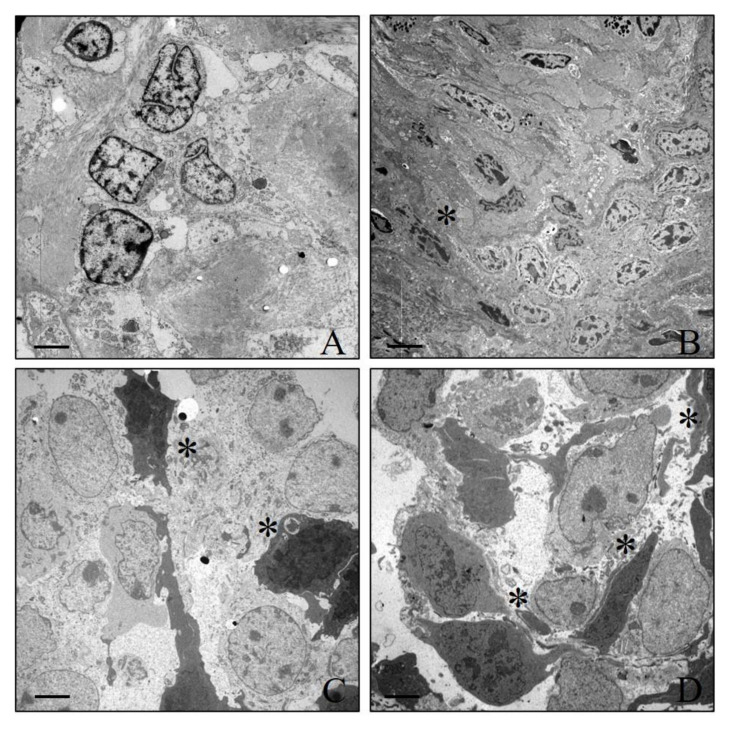
Electron microscopy investigation of xenograft tumours. (**A**,**B**) LNCAP xenograft tumour mass shows a heterogenic cell population characterized by well-differentiated prostate cancer cells and some mesenchymal-like cells (asterisks). (**C**,**D**) Images show a PC3 xenograft tumour characterized by numerous mesenchymal-like cells (asterisks). Scale bars (**A**) 5 µm, (**B**) 10 µm, (**C**) 5 µm, (**D**) 5 µm.

**Table 1 jcm-10-04850-t001:** Measurement by activity calibrator.

	30 Min		60 Min		120 Min	
	mean (MBq)	SD	mean (MBq)	SD	mean (MBq)	SD
LNCAP	0.637	0.11	0.703	0.57	0.772	0.98
PC3	0.234	0.08	0.185	0.12	0.112	0.10
Bowel	0.277	0.12	0.197	0.09	0.111	0.16
Heart	0.237	0.13	0.179	0.09	0.118	0.13
Liver	0.267	0.08	0.186	0.12	0.170	0.11
Kidney	0.351	0.15	0.300	0.11	0.110	0.12

**Table 2 jcm-10-04850-t002:** Radioactivity detection by digital autoradiography system.

	30 Min		60 Min		120 Min	
	mean (DLU)	SD	mean (DLU)	SD	mean (DLU)	SD
LNCAP	32,866	1040	39,780	3047	64,033	9623
PC3	3036	164	149.3	93.75	1	0.32
Bowel	1655	472.1	152	14	52	9.07
Heart	2455	77.7	1513	240	0.33	0.04
Liver	2285	58.6	275.7	68.09	0.32	0.04
Kidney	9856	721	1080	40.9	0.23	0.04

## Data Availability

Data will be provided on request.

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
