# Peer review of "Novel Biological and Molecular Characterization in Radiopharmaceutical Preclinical Design"

_jcm, 2021, doi:10.3390/jcm10214850_

Round 1

Reviewer 1 Report

Interesting approach to small animal imaging.

Has there been any consideration to compare this approach with a live animal in a clinical scanner ?

A better way for this whole body imaging may be with whole body slices possibly giving much better resolution to delineate more precise localisation of the radiotracer. This can be performed after the whole body imaging to compare radiotracer localisation of the proposed technique.

Author Response

Ref: jcm-1410052.

"Novel Biological and molecular characterization in radiopharmaceutical preclinical design"

Submitted to: Journal of Clinical Medicine

Before we begin the point by point review of the list of concerns, we would like to thank the Reviewer for their comments on how to improve the manuscript, which has been revised accordingly, as well as the Editors for calling for a new submission of an improved version of our manuscript.

Reply to Reviewer 1

Interesting approach to small animal imaging.

Reply: we would like to thank the Reviewer for expressing interest in our work, and for their availability to review our manuscript.

Has there been any consideration to compare this approach with a live animal in a clinical scanner ?

Reply: Thanks for this point out. In the manuscript we compared our approach with the live micro_PET analysis as follow:

“Even though the distribution of radioactivity evaluated on whole mouse body by digital autoradiography system cannot have the same sensitivity of micro-PET investigation, it allows an excellent space-time assessment of the biodistribution of a radiopharmaceutical. In particular, in this study it was possible to follow the biodistribution of a PSMA inhibitor at three different time points observing a progressive increase of radioactivity in the PSMA positive tumour area.

If compared with Micro-PET investigations, the main limitation of the use of digital autoradiography system equipped with SR screen for biodistribution studies is the impossibility to perform the analysis on live animals. In fact, the autoradiographic investigation need of a non-dynamic system to obtain an high resolution image of the radioactivity into the tissues. Conversely, micro-PET investigations can also provide physiological and pharmacological processes quantitation by dynamic acquisitions of fast kinetic data [29,30]. However, the use of micro-PET devices does not increase the animal’s welfare. Indeed, according to the D.L. March 4, 2014, No. 26; directive 2010/63/EU of the European parliament and council “Guide for the Care and Use of Laboratory Animals, United States National Research Council, 2011” animals must be sacrificed at the end of the experimental phase both using micro-PETs and this design. The difference only concerns the possibility to perform dynamic acquisitions. Micro-PET devices are extremely sensitive, having the capability to detect until a picomolar level but with PET radiopharmaceuticals only. In fact, therapeutic radiopharmaceuticals cannot detect by micro-PET limiting the possibility to perform accurate theragnostic investigations [31,32].

From our point of view, a further possible weakness for micro-PET investigations is the low spatial resolution offered by dedicated devices. This limitation makes difficult to perform accurate comparison analysis between the radiopharmaceutical uptake and microscopic characteristics of tissues. In fact, it is impossible to overlap the 3D micro-PET images with the histological characteristics of investigated tissues.”

A better way for this whole body imaging may be with whole body slices possibly giving much better resolution to delineate more precise localisation of the radiotracer. This can be performed after the whole body imaging to compare radiotracer localisation of the proposed technique.

Reply: we would like to thank the Reviewer for this interesting consideration. We agree that the analysis of whole-body slices can significantly improve the resolution to delineate more precise localisation of the radiotracer. However, the aim of this study was demonstrated the capability of digital autoradiography system equipped with SR screen is performed preliminary evaluation of a biodistribution of a radiolabeled molecule. In the new version of our manuscript, we added a brief discussion of this.

“The use of digital autoradiography system equipped with SR screen can represent an extraordinary and less expensive alternative for all research groups to perform preliminary and multidisciplinary imaging investigations, with all available radiopharmaceuticals. It is important to note that the methodology here proposed could be also associated with the analysis of whole-body slices performed by micro-PET investigations to delineate more precise localization of the radiotracer.”

Reviewer 2 Report

Authors submitted their work regarding a novel device for prelclinical characterization of new radiopharmaceuticals. My concerns are below:

Some items are missing for reproducibility of experiments :

General comments:

  • 18F-PSMA: in house production? If yes, radiolabeling and QC should be reported. If not the supplier should be mentioned
  • For animal experiments, the reference of the ethics committee must be reported
  • Activities should not been reported in uCi but in kBq or MBq
  • Immunofluorescence images of PSMA and Ki-67 in LNCaP and PC3 cells should be presented
  • An image of the newly developed system would be appreciated

Major points are below:

  • Authors claimed that uptake of 18F-PSMA is specific but blocking experiments are missing
  • Authors presented a new device. Therefore, it would be interesting to demonstrate that several radioisotopes can be used. The use of solely 18F is a major drawback. The use of several radionuclides such as diagnostic emitters (gamma, X) and/or therapeutic ones (β, α) would show to the reader the versatility of the system. Moreover, qualification of the device is mandatory, in term of LOD, LOQ, sensivity, resolutionetc. Overall, this paper is “only” an example of the possibilities offered by this system.
  • Other similar devices have been developed and used in the literature. A comparison would be appreciated

For these reason I suggest a resubmission of the work if authors can provide a more comprehensive study of this new device

Author Response

Reply to Reviewer 2

Authors submitted their work regarding a novel device for prelclinical characterization of new radiopharmaceuticals.

Reply: we would like to thank the Reviewer for his/her availability to review our manuscript. We hope that the modified version of our manuscript can meet his/her comments.

18F-PSMA: in house production? If yes, radiolabeling and QC should be reported. If not the supplier should be mentioned

Reply:  Thanks for this point out. The 18F-PSMA used in this study was provided by ITEL Telecomunicazioni S.r.l. Currently, we do not provide more information about this molecule. We specified this in the Acknowledgments section.

For animal experiments, the reference of the ethics committee must be reported

Reply:  we added this “Experimental protocols were approved by the Animal Care and Use Committee at the institution involved in this study and by the Italian Ministry of Health (authorizations N. 951/2020-PR).” In the “Institutional Review Board Statement” section.

Activities should not been reported in uCi but in kBq or MBq

Reply:  Thanks for this point out. We modified our data according to the reviewer suggestion.

Immunofluorescence images of PSMA and Ki-67 in LNCaP and PC3 cells should be presented

An image of the newly developed system would be appreciated

Reply:  in the new version of our manuscript, we added immunofluorescence images of PSMA and Ki-67 in LNCaP and PC3 cells, in supplementary material, and an image of the autoradiographic system here described (New Figure 1).

Authors claimed that uptake of 18F-PSMA is specific but blocking experiments are missing

Reply:  thanks for this point out. The aim of this study was only demonstrated the possibility to perform preliminary biodistribution studies with the digital autoradiography system equipped with SR screen. Therefore, in the new version of our manuscript we deleted every reference concerning the specificity of the investigated 18F-PSMA.

Authors presented a new device. Therefore, it would be interesting to demonstrate that several radioisotopes can be used. The use of solely 18F is a major drawback. The use of several radionuclides such as diagnostic emitters (gamma, X) and/or therapeutic ones (β, α) would show to the reader the versatility of the system. Moreover, qualification of the device is mandatory, in term of LOD, LOQ, sensivity, resolutionetc. Overall, this paper is “only” an example of the possibilities offered by this system.

Reply:  thanks for this point out. We agree with all considerations of the reviewer. However, this is the first preliminary investigation that demonstrated the possibility to perform biodistribution studies by using the proposed digital autoradiography system. In addition, the characteristics of the SR screen make this device suitable for biodistribution studies of other radionuclides such as diagnostic emitters (gamma, X) and/or therapeutic ones (β, α). The detailed characteristics of the system are reported in the manuscript.

“Digital autoradiography system is a very versatility and sensitivity device for radioisotope imaging, replacing film autoradiography [27]. This system has been designed for a great variety of applications such as analysis of purity for radiopharmaceuticals, nucleotide metabolism studies, in vitro imaging of tissue sections and also gene and protein ex-pression studies [27]. In fact, it can image and quantify activity distribution of different radionuclides (photons-, β- and α-particles emitting).

SR phosphor screen is a flexible support film formulated with the finest grade of barium fluorobromide containing trace of bivalent europium (BaFBr/Eu2+) phosphor crystal, which acts as bioluminescence center, to provide the best resolution. When the screen is exposed to radioactive sample, the energy of radioisotope ionizes the Eu+3 to Eu2+, liberating electrons which are trapped in the bromine vacancies [28]. Subsequently, the ex-posed SR screen, wrapped around the carousel of the photometer reading device, is scanned by a focused red light laser beam (633nm); the laser-stimulated luminescence re-leases blue light photons (390nm) which are detected by a photo-multiplier tube (PMT) and converted to electrical signals expressed as DLU. The SR screen was scanned in a few minutes to create a high-resolution digitized image of the locations and intensity of the radioactive in the sample that is quantified by OptiQuantTM image analysis software and stored for future reference.”

We also agree that new study focused on the biodistribution of diagnostic emitters (gamma, X) and/or therapeutic radionuclides are needed.

“However, it has not escaped our notice the needed to perform further studies focused on the biodistribution of diagnostic emitters (gamma, X) and/or therapeutic radionuclides in order to complete the validation of the digital autoradiography system here described.”

Other similar devices have been developed and used in the literature. A comparison would be appreciated

Reply:  thanks for this point out. In the revision form of our manuscript, we added comments about the clinical relevance of our study also better specifying the rationale of the study.

“For several years, the in-situ detection of radiolabeled molecules has been performed by using film or film emulsion (conventional autoradiographic analysis). Despite the spatial resolution obtained with these devices is very good, the sensitivity of film for the low activity levels is poor since the low x-ray/β particle detection efficiency. According to this, film autoradiograph frequently must take several days to produce a satisfactory image. In addition, the limited dynamic range of film can cause under- or over-exposure of parts of the image. Therefore, better autoradiography systems based on digital position-sensitive detectors have been developed. Among these, the most sensitive are phosphor imaging plates [27], multiwire proportional chambers [28], scintillating optical fibres [29], micro-channel plates [30], silicon strip detectors [31], silicon or gallium arsenide pixel detectors [32]. Moreover, in the last years extremely sensitive digital autoradiographs have been developed both for quality control and in vivo research.”

Round 2

Reviewer 1 Report

An interesting way of presenting gross small animal imaging without the need for dedicated small animal imaging equipment and facilities.

Reviewer 2 Report

I would like to thank author for their prompt revision of the article. I understand well that it is only a preliminary study, but I cannot accept this article which is too preliminary. I would congratulate authors for their effort, and I encourage them to pursue their study that will be published in a good journal for sure once they performed all control experiments and qualification of the device. Good luck!